# Global Energy Transformation and the Impacts of Systematic Energy Change Policy on Climate Change Mitigation

**Hakan Güneş [1,*], Hamis Miraji Ally Simba [2], Haydar Karadağ [3] and Mustafa Şit [4]**

[1] Accounting and Taxation Program, Bartin University, Bartın 74100, Turkey
[2] Department of Economics and Finance, Istanbul Gelisim University, Istanbul 34000, Turkey; hamisimiraji2020@gmail.com
[3] Department of Economics, Recep Tayyip Erdogan University, Rize 53100, Turkey; haydar.karadag@erdogan.edu.tr
[4] School of Tourism and Hotel Management, Harran University, Sanlıurfa 63100, Turkey; msit@harran.edu.tr
[*] Correspondence: hakangunes@bartin.edu.tr

**Abstract:** This study aims to evaluate the effect of global energy transformation and systematic energy change on climate change. The model is constructed from dynamic panel data which comprises 26 world regions from the World Database Indicators (WDIs), International Energy Atomic (IEA), and International Monetary Fund (IMF), with a span from 2005 to 2022. The Generalized system Method of Moment (sys-GMM) and pooled OLS and random effect models have been used to empirically evaluate the linked effect of global transformation and systematic change on climate change. The sys-GMM approach is used to control the endogeneity of the lagged dependent variable when there is an association between the exogenous variable and the error term. Furthermore, it omits variable bias, measurement errors in the estimation, and unobserved panel heterogeneity. The econometric applications allow us to quantify the direct effect of global transformation and systematic change on climate change. The empirical analysis revealed that renewable energy, alternative energy, technology and innovation, and financial climate have a negative effect on climate change. It means that increasing consumption of the transformation energies leads to reducing the effect of climate change. However, fossil energy is statistically significant and positively affects climate change. Increasing the consumption of fossil energy raises the effect of climate change**.** There is a global need for massive decarbonization infrastructure that will help minimize the global warming that leads to climate change. Policies that take an endogenous approach through global transformation and systematic change should be implemented to reduce the effect of climate change. The policy should reduce the consumption of non-renewable energy and increase the consumption of renewable energy.

**Keywords:** global energy transformations; systematic energy changes; climate changes; sys-GMM

## 1. Introduction

Climate change is a serious issue that should urgently be solved because it causes calamities and catastrophes that affect the ecological and biodiversity of the Earth. It is far more rapid and dangerous than thought before. Aside from that, it destroys the nature of biodiversity, deteriorates human health, and decelerates socioeconomic development [1,2]. However, human economic development has contributed to the existence of climate change, especially after the industrial revolution, by burning fossil fuel, heating coal, and using gas for industrial energy consumption, leading to excess carbon dioxide and greenhouse gas emissions which have tremendous effects on living things [3]. The major development of energy production, like oil and gas extraction and exploration for human consumption, somehow contributes to climate change by producing large quantities of greenhouse gas and carbon dioxide gas in the atmosphere [4]. Multiple effects have been

recorded globally due to the increase in global warming, which changes the nature of biodiversity and habitat of the Earth. Furthermore, global warming causes the melting of glaciers in the Antarctic and Greenland ice sheets, a higher frequency of severe hurricanes, greater severity of droughts, and forest fires [5]. The melting of ice glaciers in different parts of the Earth resulted in increases in ocean and sea depths, leading to heavy rainfall and heavy windstorms [6]. The increase in global warming not only causes climate change but also affects the ecological system as a result of the distortion of water, land, forests, wildlife, and fisheries [7]. The increase in climate change has been attributed to different factors such as greenhouse gas emissions, fuel gas combustions, and carbon dioxide gas emissions from industries, transportation systems, and other energy sectors [8]. For instance, the consumption of non-renewable energy from giant industries and transportation emits carbon dioxide and greenhouse gases into the atmosphere, which triggers climate change [9]. Generally, global warming raises global temperatures, which has adverse effects on cultural, social, and economic development [10]. Thus, different meetings and conferences have been held to protect the Earth from climate change. In addition to the thorough scientific investigation on the negative impact of the global rise in temperature by 2.0 °C that causes climate change, the world's countries continue to implement effective policies to mitigate climate change [11]. For example, the COP 26 climate change conference in Glasgow is a global initiative to campaign for climate change with the aim of lowering global warming from 1.5 to 0.0 °C. Figure 1 illustrates the transition of global energy transformation from the Kyoto protocol of 1997, the Paris Agreement and SDGs, and later the COP 26 Glasgow and COP 27 in Egypt with the aim of reducing greenhouse and carbon dioxide gas emissions that cause climate change [12–14].

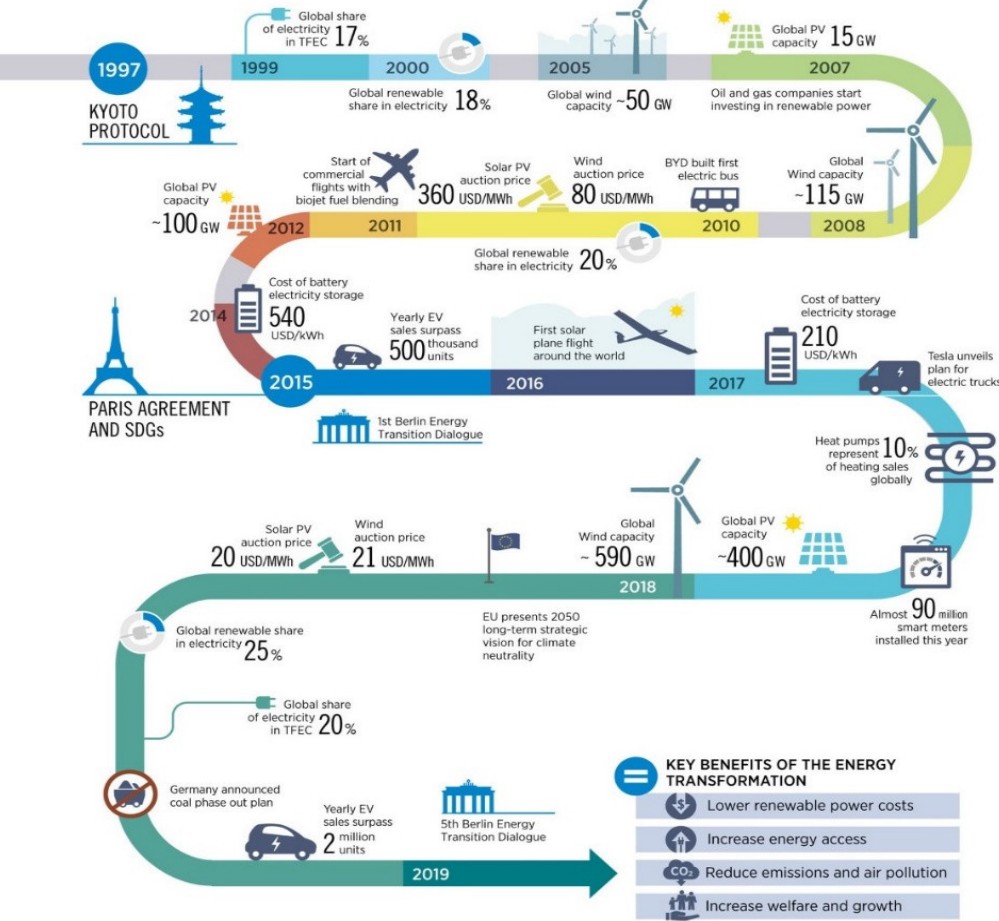

**Figure 1.** Global energy transformation.

## 2. Theoretical Background

Different approaches have been put forward to discuss the effects of global energy transformation and systematic energy changes on climate change [14]. Bui et al., have stated that global energy transformation is significant and vital to global climate policy because it is a multifaceted transformation that involves socioeconomic policies, technological advancement, and institutional drivers to mitigate the effect of climate change [15]. Panwar et al., argued that regarding recent and post-climate change mitigation, renewable energy is the optimal clean energy source that is environmentally friendly and emits the minimum amount of greenhouse and carbon dioxide gas as the means of maintaining sustainable energy development toward climate change [16,17].

Panwar et al., also claim that technology from renewable energy is an exceptional opportunity for climate change mitigation in a way that it reduces global warming by producing a low rate of greenhouse and carbon dioxide gas emissions via replacing the transformation of energy sources [17].

The urgency of climate change mitigation comes with an opportunity for global energy consumption to transform from non-renewable energy to renewable energy toward sustainable energy development. The postulates of renewable energy for optimization in advanced modeling sectors have the advantage of negative energy emissions effects against climate change. The renewable energy system for mitigating climate change through electric vehicles, marine vehicles using hydrogen propulsion, the accurate prediction of long-term wind speed and the monitoring of innovations in water quality includes renewable energy technologies to produce less greenhouse gases and carbon dioxide gas in the atmosphere [18].

According to Hassan et al., alternative energy is the substitute for fossil fuel energy, which emits large amounts of carbon dioxide and greenhouse gas into the atmosphere. Investment in alternative energy should be part of strategies to mitigate climate change. The investment in alternative energy deployment minimizes decarbonization. Thematically, the improvement of solar panels, nuclear energy, geothermal and biofuel production from biomass, green hydrogen innovations, and heat transfer for sustainable combustions reduces the effect of climate change. The upscaling widespread solutions of climate change mitigation requires a strong and effective approach that includes integrated coordination and synergistic strategy which is, in a way, the only chance to protecting the world against climate change. Alternative energy is a climate policy that encompasses the set of minimization of environmental impact hazards, economic affordability, and energy security for a long-term plan of decarbonization in 2050, and of course, it includes parameters such as financing standards and marketing energy design. However, the adoption of alternative energy for the mitigation of climate change is not possible without the deployment of energy development systems [19].

Hook et al., explained that the post-scenario anthropogenic climate change discipline revealed the increase in carbon dioxide and greenhouse gas in the atmosphere as the result of consuming large amounts of fossil fuel. Meanwhile, the depletion of fossil energy is also identified as a global energy crisis. Therefore, the limits of consumption of fossil energy will set the household consumption ability that affects climate change [20]. Gardiner et al., state that fossil fuel energy benefits domestic consumption by providing abundant as well as cheap and versatile energy access. Whenever it happens, fossil energy is unprecedented and dramatically improves life expectancy as well as reduces the mortality rate by enhancing the development indicators such as health services and the right to education and reducing the unemployment gap by increasing the volume of energy consumption in the community. If frontiers economies fail to access an abundance of fossil fuels, the development indicators might be less reliable, and poverty risks might increase. Since the impact of the emission of carbon dioxide and greenhouse gas, emissions continue to exacerbate the effects of global warming, and fossil fuels produce excess carbon dioxide gas and greenhouse gas into the atmosphere; thus, fossil fuels energy has become a great agent to contribute to the rising global temperature. [21].

Gans et al., state that tighter limits on carbon dioxide and greenhouse gas emissions reduce fossil fuel use, which in turn reduces fossil fuel efficiency. Such a policy will stimulate the demand for innovation that improves alternative energy and renewable energy innovation, which leads to a decrease in carbon dioxide and greenhouse gas emissions. Only innovation technology that abates carbon dioxide emissions has a negative impact on climate change [22]. Ma et al., have conducted different studies indicating that innovation technology is leading to minimizing the effect of climate change, and a similar result has been reported by Kihombo et al. [23,24]. On the contrary, the research conducted by Mughal et al., revealed the insignificant positive correlation between innovative technology and climate change [25]. However, Godil et al., have found that innovation technology and renewable energy both have a detrimental effect on carbon dioxide emissions related to transportation [26].

The United Nations Framework Convention on Climate Change (UNFCCC) depicts that climate finance ought to be managed by a private or government financial institution that has the institution investment to support the climate change mitigation and adaptation of all features with the intention of decarbonization processing. It is an investment where the joint government collaborates with private institutions to undertake the transition of the global energy level so as to lower carbon path emissions and rebuild global resilience. Zhang et al., state that climate finance energy needs huge investments in low-carbon and greenhouse emissions infrastructure; however, the funds that have been relocated for the expansion of climate finance globally are inadequate [27]. Weikmans et al., emphasize the need for strict rules and regulations within the corporate structure for the initiation, improvement, and accountability of climate finance fund projects. The climate finance institutions, via green climate finance funds, should decarbonize the energy sectors and initiate the construction of green buildings to reduce the effect of climate change [28].

Acaroglu et al., investigated the effect of renewable energy consumption on climate change mitigation. The study was conducted by using data from the World Bank 1980–2019 and employing autoregressive distributed lag and Toda–Yamamoto to find the causal relationship between renewable energy consumption, economic growth, and climate change. The outcome indicates a negative unidirectional causality that runs from renewable energy consumption to climate change in both the short-run and long-run periods [29]. Russia et al., investigated the effect of renewable energy consumption on climate change with the objective of observing the energy supply chain of decarbonization output in future trends. The study, which used the variables wind speed, temperature, humidity, and solar irradiation, extracted the data from the global climate model with a sample size of 43 climate energy systems and employed the methods of a global climate model (GCM), regional climate (RCM), Matlab, and Python. The findings indicate that the significant largest power of renewable energy was estimated in the long term while non-significant climate change was estimated in the short term. The highest access variability of renewable energy was found in wind power and followed by hydro-electrical power generation. Both were found in the long-term periods. Additionally, the decarbonization variability was invested in wind power as the source to abate the effect of climate change [30].

Banga et al., argue that the tourism sector has been accused of being the major contributor to global warming due to the size of its industry, which subsequent to high energy consumption, mostly emits carbon dioxide. However, the current study, which includes the dynamic-GMM for 38 OECD countries from 2008 to 2019, alluded to the fact that the tourism sector has zero carbon dioxide emissions. However, the result shows that the tourism links of the OECD countries are non-significant in terms of greenhouse gas emissions. Therefore, using renewable energy sources instead of non-renewable energy consumption should continue. Previously, the consumption of renewable energy stimulated the global energy carbon and greenhouse emissions to attain carbon neutrality, which is the main objective of the United Nations [31]. Hao et al., explored the impact of alternative energy and economic growth on climate change for 105 countries from 1990 to

2019 by using the construction of a panel vector autoregressive (PVAR) model and a generalized method of moments (GMM). The results find that higher and upper-middle-income states of industrialized countries have significant and positive effects on carbon dioxide and greenhouse gas emissions. Moreover, the results show that renewable energy consumption reduces carbon dioxide and greenhouse gas emissions into the atmosphere [32].

Mirziyoyeya et al., investigated the effect of alternative energy consumption on climate change from 2005 to 2015. The research employed fixed effect regressions and two-step sys-GMM for estimations. The estimated results revealed that alternative energy consumption has a negative and significant effect on climate change. The results showed that increasing alternative energy consumption decreases climate change by about 0.98% [33]. Naseem et al., investigated the effect of alternative energy, agriculture and economic growth on climate change based on SAARC regions using the annual cross-section data from 2000 to 2017. The study used the fixed effect and two-step sys-GMM model to check the robustness of the variables. The results revealed that renewable energy and agriculture have a negative and significant effect on climate change. Moreover, decarbonizations are essential drivers for climate change mitigation. Therefore, regional cooperation may accelerate the improvement of alternative energy consumption due to lowering the effect of climate change [34].

Irfan et al., analyzed the causality between nuclear energy consumption and climate change based on the data regarding developing countries from 1980 to 2020. The research used the BRW method to examine the potential variation of time in causality in time series data. The results indicated that there is a negative unidirectional causality that runs from nuclear energy consumption to climate change. There is a positive causal direction from nuclear energy consumption to climate change in the USA and Germany, while a negative unidirectional causality from nuclear energy consumption to climate change was detected in Canada and France. A positive causality shows that nuclear energy consumption weakens environmental protections, which leads to an increase in climate change [35]. Cheng et al., investigated technological innovation and the mitigation of carbon dioxide emissions using evidence from China. The study first applied the condition mean (CM) to investigate the result. Then, quantile regression was employed to examine the importance of the heterogenous effect comprehensively. An extended STIRPAT model has also been used to examine the impact of renewable energy technology innovation and fossil technology innovation on carbon emissions. The results revealed that renewable energy technology innovation has a significant positive effect on climate change intensity in lower quantile areas and a negative effect on the higher quantile areas. Additionally, fossil energy technology innovation shows the negative and positive aspects of carbon emission on both lower quantile intensity and higher quantile intensity, respectively [36].

Elheddad et al., investigated the impact of foreign direct investment (FDI) toward non-renewable and renewable energy in Bangladesh. The study set the time series regressions, non-parametric (quantile regressions) and parametric (GMM. IV estimations) models to investigate the FDI, non-renewable and renewable energy on climate change. The results from the investigations revealed that the FDI inflows cause more climate change in the Bangladesh economy. Moreover, the usage of non-renewable energy (fossil fuel) increases the emissions of carbon dioxide as well as the pollution of the atmosphere. Moreover, the results indicate that the FDI debated renewable energy consumption, and the negative relationship found between FDI on renewable energy is greater than the positive effect of FDI on carbon dioxide and greenhouse gas emissions [37]. Li et al., investigated the exogenous status of innovation and renewable energy against endogenous greenhouse gases using 2000–2019 data in both OECD and non-OECD countries. The research employed the sys-GMM and simultaneous equations and models. The results revealed that renewable energy in non-OECD countries was significant and stimulated the growth of greenhouse gas emissions, and the contrary was found for OECD countries. Moreover, renewable energy abates greenhouse gas caused by climate change, precisely for OECD countries. In fact, the study revealed that innovation technology input

has a significant effect on increasing renewable energy and mitigating the greenhouse gas emissions as well as promoting green economic growth in OECD countries. Thus innovation technology has an impact on both OECD and non-OECD countries regarding climate change in the short term, medium term, and long term [38].

Blanco et al., investigated the effect of climate change using the variables of innovation, technology development and energy transfer. The research used the data collected through sampling procedures of 516 companies from manufacturing industries in Pakistan. The study used structural equation modeling (SEM) and artificial natural network (ANN) methodologies. The research results showed that all the variables of SEM and ANN of all the research model constructs are related to the greenhouse and carbon dioxide gas integrated from innovation costs and facilitating conditions. Social influence, hedonic behavior, and effort expectancy predict the green behavioral intention (GBI) of the decarbonization process; the results indicate the differentiation in the emission of greenhouse and carbon dioxide gas among the small, medium and large enterprises in Pakistan. In fact, the results show that all the integrated constructs have important values for measuring greenhouse gas and carbon dioxide emissions [39]. Zhang et al., investigated green finance to counteract climate change mitigation. The study investigated the relationship between low-carbon infrastructure investments, financing, and climate change mitigation. However, a financial resource shortage was detected. The investigation runs over the period of 2008–2018 for G-8 economies. The output of the results was examined through digital finance and green finance on environmental protection using a quantile regression model. The results of the analyses revealed that the carbon dioxide emissions were reduced by green finance, renewable energy investment, and technological innovation. In contrast, non-renewable energy consumption was found to increase carbon dioxide and greenhouse gases. Doku et al. (2022) [40] investigated bilateral finance and food security in developing countries. The study examined the effect of green technology innovation and green investment on reducing the effect of carbon dioxide emissions. It used the data from 1995 to 2019 and invites the cross-sectional dependence structure break and slope heterogeneity, which includes the cointegration test and autoregressive distributed large (ARDL), Banerjee, and Carrion-I Silvestre unit root. The study revealed the negative relationship between green technology innovation (GINV) and green financing (GFIN) on the emission of carbon dioxide gas in G-7. The review of the literature shows the number of gaps in such a way that most of the articles did not examine the effect of global energy in the form of global energy transformation and systematic energy change [40]. However, this study examines the effect of climate change in the context of proxies of global energy transformation and proxies of systematic energy change and how they affect climate change.

The conceptual framework of this study is based on global energy transformation, systematic energy changes, and climate change mitigation. The global energy transformations sections consist of renewable energy, alternative energy, and fossil energy, while the systematic energy change includes technology innovation as well as climate finance for climate change mitigation (Figure 2):

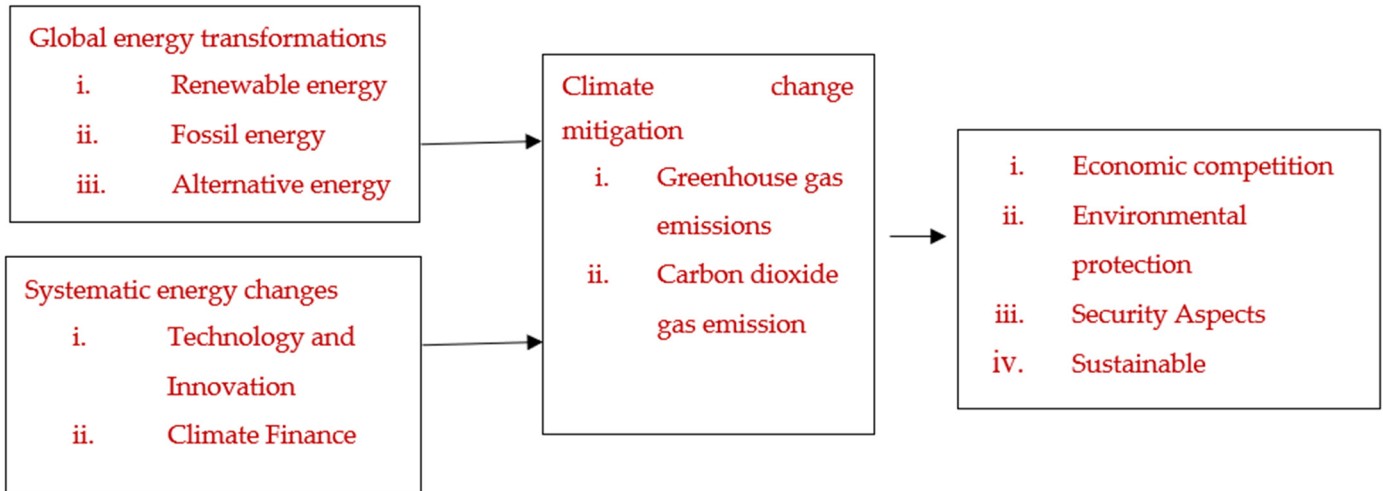

**Figure 2.** Conceptual framework.

The objective of the study is to evaluate the impact of global energy transformation and systematic energy change on climate change mitigation. Although climate change meetings and conferences have been held regarding climate change mitigation, states and non-state institutions concerned with climate change have pledged further efforts toward climate change mitigation. Such efforts in climate change mitigation enable existing and future researchers to explore new dimensions that acquire deeper perspectives by providing a broader assessment of climate change mitigation. There are some remarkable studies reviewing the climate change mitigation literature, but none of them include the effect of proxies of global energy transformation change and systematic energy change on climate change mitigation. Moreover, the empirical analysis of this study is based on the comprehensive dataset covering information related to global energy transformation, systematic change, and climate change. In addition to the thorough scientific research on the harmful effects of climate change, one of the most important reasons this issue continues to be important is that global warming still exceeds 1.5 °C, accelerating climate change. Therefore, the climate change mitigation policy will require a manageable and functional system to be translated into energy transformation and systematic energy policy development. In addition to that, the study validates both by testing whether the global energy transformation and systematic energy change have a negative or positive impact on climate change mitigation. Therefore, this study is expected to contribute to the literature by identifying the negative impact of global energy transformation and systematic change on climate change mitigation. The paper starts with a description of the global energy transformation, systematic energy change, and climate change in section one. Then, it discusses the thoughts and arguments from different scholars related to climate change in Section Two. Section Three is related to the data analysis and methodology. Section Four presents the results and discussion, and the last section includes a conclusion and policy recommendation**.**

## 3. Materials and Methods

The aim of this study is to evaluate the effect of global energy transformation and systematic energy change on climate change. The model is constructed from balanced dynamic panel data, which comprises 26 world regions as represented in Appendix A from the World Database Indicators (WDIs), International Energy Atomic (IEA), and International Monetary Fund (IMF), which spans from 2005 to 2022. The reason for choosing these 26 regions is to assess the impact of energy conversion and systematic energy change to mitigate climate change on a global scale. Due to the difficulty of accessing the data quality related to climate change, this study investigates the effect of climate change by using

balanced panel data using the 26 world economic regions. The data presentation for investigating the global effect of climate change is based on the demographic and production rate of climate change and greenhouse gas emissions in the atmosphere.

This study addresses the key policies of global energy transformation and systematic energy change regarding climate change. The global energy change is the process that transmits energy from non-renewable energy use to renewable energy to decarbonize greenhouse gas and carbon dioxide gas. The global energy transformation includes renewable energy, alternative energy, and fossil energy [14]. The global energy transformation is associated with transitional sustainable energies, which depend on climate change mitigation policy technology, energy efficiency for electric power generation, and the generation of renewable energies for domestic and industrial consumption. The systematic energy change includes technologies and innovation as well as climate finance. Both global energy transformation and systematic energy change are policies of climate change mitigation that have been introduced to minimize the effect of climate change. However, the global energy transformation is complex and difficult to be implemented due to the fact that it needs innovation and technology as well as financial assistance from the existing intensive fuel energy use to support renewable consumption.

### 3.1. System GMM

For this purpose, this study uses the sys-GMM, random effect (REM) and pooled OLS to examine the effect of global energy transformation and systematic energy change on climate change. The sys-GMM regression technique was introduced as an appropriate analysis method for investigating the effect of global energy transformation and systematic energy change toward climate change. It revealed that sometimes, the panel data analysis may cause endogeneity problems. Therefore, the sys-GMM technique has been introduced to overcome the endogeneity problem. The GMM technic is categorized into two options, difference GMM and system GMM, but the sys-GMM is preferable compared to difference GMM because the sys-GMM has been proven to correct biases due to unobserved, global regions heterogeneity, omissions of variables, and errors due to the measurement estimation and actual tendency of endogeneity that affect the output of the results [41]. Furthermore, the modeling strategy, which is more dynamic, enables the control of persistence, which has persistent behavioral effects. The persistence can be checked through the correlation between the endogeneity variable and its corresponding first lag. The sys-GMM technique assumes endogeneity by controlling the heterogeneity with time-invariant omitted variables. Therefore, the cross-country variation is normally controlled in the regression equations. Furthermore, Blundell and Bond (1998) [42] argued that the sys-GMM estimator corrects the biases which have been associated with the difference estimator. The sys-GMM is more concerned with the panel data in which the number of the cross-section (N) is greater than the number of the periods (year)T; as this is the case, there are some endogenous variables that are not strictly embedded to the exogenous variables. Moreover, it is based on the presence of heteroskedasticity and autocorrelation within the data. The other section is based on the orthogonal deviations, which are different alternatively from the original transformation used in the traditional GMM approach and that prepared by Arellano and Bond (1995), which was adopted in this investigation [41,42]. The dummy variables have been included because the dummy variables capture the time-variant specific effect. In fact, the dummy variables not only reflect the assumption of no correlation across the global regions but also demonstrates the specific function of reducing the extent of serial correlation among the data and idiosyncratic terms. Moreover, it improves the robustness of the estimation of the results. [43]. However, ignoring the effect of proliferation or over-identification of instruments leads to bias toward the GMM approach, over-fitting of endogenous variables, and weakening of the Sargan/Hansen test. As the rule of thumb states, the number of the instrument should not be higher or greater than the number of periods of time cross-sections [44]. However, Rodman suggests that to avoid the proliferation of instruments,

the number of instruments should be smaller than the number of the groups. In this context, the two-step method of sys-GMM is robust to the heteroskedasticity autocorrelation, which removes the standard error bias. The instrument was set to lag 2 to make the results more robust. Post-estimations for sys-GMM have been introduced to examine the accuracy of the sys-GMM estimates, especially the coefficients of the lagged regress relative to pooled OLS and the random effect model. Thus, the sys-GMM uses the lagged 2 to estimate the results. The sys-GMM is represented as follows:

$$CO_{2it} = \beta_{it} + CO_{2i(t-1)} + \beta_1 RE_{it} + \beta_2 AE_{it} + \beta_3 FE_{it} + \beta_4 (T\&I)_{it} + \beta_5 CF_{it} + \lambda_t + \eta_i + u_{it} \tag{1}$$

$$GHG\text{-}CO_{2(it)} = \beta_{it} + GHG\text{-}CO_{2(t-1)} + \beta_1 RE_{it} + \beta_2 AE_{it} + \beta_3 FE_{it} + \beta_4 (T\&I)_{it} + \beta_5 CF_{it} + \lambda_t + \eta_i + u_{it} \tag{2}$$

The dimensions of cross-section and time-series descriptions are represented by i and t subscripts. Climate change $CO_2$ and greenhouse gas emissions $GHG\text{-}CO_2$ are both endogenous. At the same time, renewable energy (RE), alternative energy (AE), and fossil fuel energy (FE) as the proxies of global energy transformation energy and technology innovation (T&I) and climate finance (CF), the proxies of systematic energy change, are acting as exogenous variables. $\beta$ represents the coefficient parameters of the respected variables. $\lambda_t$ is the time-specific term, $\eta_i$ is the global regional-specific term, and $u_{it}$ is the error term/disturbance term that captures the omitted variables. i is the cross-sectional data. t represents the time series of the data. As depicted earlier, some variables are endogenous in nature; therefore, to address the endogeneity problems that may be present in Equations (3) and (4), we have to apply the instrumental variables regression model based on the GMM technique. Therefore, we validate the instruments by adopting Rodman's [45] via the imposition of lags and collapse to reduce the proliferation of instruments.

### 3.2. Pooled OLS

Therefore, from Equations (3) and (4), we can estimate the pooled OLS. The analysis included the pooled OLS model by considering the effect of the proxy variables of global energy transformation and systematic energy change on climate change. The pooled OLS had the time-constant attributes of individuals that are not correlated with individual regressors. It revealed the unbiased and consistent estimators of parameters even if the time-constant attributes were present.

$$CO_{2it} = \beta_{it} + \beta_1 RE_{it} + \beta_2 AE_{it} + \beta_3 FE_{it} + \beta_4 (T\&I)_{it} + \beta_5 CF_{it} + u_{it} \tag{3}$$

$$GHG\text{-}CO_{2(it)} = \beta_{it} + \beta_1 RE_{it} + \beta_2 AE_{it} + \beta_3 FE_{it} + \beta_4 (T\&I)_{it} + \beta_5 CF_{it} + u_{it} \tag{4}$$

### 3.3. Random Effect Models

The random effect (RE) model has been applied to assume that the unobserved heterogeneity of variables will not bias the estimated results [46]. The random effect measures the effect of a random factor that is associated with exogenous indicators [47]. Moreover, it is conducted for the robustness errors if there is a presence of heteroskedasticity in the applied data [48]. Therefore, the other advantage of applying the random effect is to eliminate heteroscedasticity.

$$CO_{2it} = \beta_1 + CO_{2i(t-1)} + \beta_1 RE_{it} + \beta_2 AE_{it} + \beta_3 FE_{it} + \beta_4 (T\&I)_{it} + \beta_5 CF_{it} + \varepsilon_i + u_{it} \tag{5}$$

$$GHG\text{-}CO_{2(it)} = \beta_1 + GHG\text{-}CO_{2(t-1)} + \beta_1 RE_{it} + \beta_2 AE_{it} + \beta_3 FE_{it} + \beta_4 (T\&I)_{it} + \beta_5 CF_{it} + \varepsilon_i + u_{it} \tag{6}$$

Equations (5) and (6) consist of two components: $\varepsilon_i$, which is the cross-section, or individual-specific, error component, and $u_{it}$, which is the combined time series and cross-section error component and is called the idiosyncratic term because it varies over cross-section as well as time.

Table 1 includes the variables obtained from the World Database Indicators, International Atomic Energy Agency, and International Monetary Fund used in this research.

**Table 1.** Variable description.

| Variables | Description | Symbol | Data Source |
|---|---|---|---|
| Global energy transformation | Climate change policy for climate change mitigation | GET | World Development Indicator |
| Systematic energy change | Climate change policy for climate change mitigation | SEC | World Development Indicator |
| Climate change $CO_2$ emission | Climate change expressed as $CO_2$ of gaseous fuel consumption (% of total) | $CO_2$ | World Development Indicator |
| Greenhouse gas emission | Greenhouse gas emission expressed as $CO_2$ of gaseous fuel consumption (% of total) | GHG_$CO_2$ | World Development Indicator |
| Renewable energy consumption | Renewable energy expressed as % of total final energy use | RE | World Development Indicator |
| Alternative energy | Alternative energy expressed as % of total energy use | AE | International Atomic Energy Agency |
| Fossil energy | Fossil energy expressed as % of total energy use | FE | International Atomic Energy Agency |
| Technology and Innovation | Technology and innovation as % of manufactured export | T&I | World Development Indicator |
| Climate finance | Climate finance as % of US dollars | CF | International Monetary Fund (IMF) |

Table 2 shows that the average mean of global renewable energy is 0.2891, while the expansion of renewable energy reached 0.8034, and the minimum is about 0. 231. This indicates that up to 2021, renewable energy consumption based on electricity increased. Moreover, the standard deviation of renewable energy from 2007 to 2021 was 0.0037. Table 2 also reports that the average mean of fossil fuel energy is about 0.1932, which indicates the decline in fossil fuel energy consumption compared to renewable energy. Moreover, fossil fuel energy has the maximum percentage of energy consumption, which is about 3.145, and the deviation of fossil fuel energy is 0.063. The decline in the average mean of fossil fuel energy consumption is caused by the transition of global energy. Alternative energy has an average mean of 0.1723, and a standard deviation of 0.0015, and its maximum value is about 0.3072. The average mean of greenhouse gas emission is about 0.3792, and its standard deviation is 0.0314, which makes the maximum value 0.8345. The average mean of global carbon emission reached 0.4897 while the maximum global carbon emission is about 5.9213, and the deviation of the carbon emission is 0.0879. The average mean of technology and innovation is 0.2325, while the maximum value of innovation-technology is about 0.1923. The average mean of climate finance is 0.1436, while the maximum value is about 1.1923. Moreover, the table illustrates descriptive statistics values which show that the skewness value is above 75% and the kurtosis values are more significant than 1.5, which means all the applicable proxy variables are free from multicollinearity.

**Table 2.** Descriptive statistics.

| Applied Variables | Obs | Mean | Median | Std.Dev | Max | Min | Skewness | Kurtosis |
|---|---|---|---|---|---|---|---|---|
| RE | 759 | 0.2891 | 0.2332 | 0.0037 | 0.8034 | 0.0131 | 0.7832 | 3.0121 |
| FE | 759 | 0.1932 | 0.1732 | 0.0063 | 3.1456 | 0.0231 | 0.8135 | 2.142 |
| AE | 759 | 0.1723 | 0.2673 | 0.0015 | 0.3072 | 0.0041 | 0.7745 | 4.1113 |
| $CO_2$-GHG | 759 | 0.3792 | 0.1234 | 0.0314 | 0.83451 | 0.0009 | 0.8934 | 3.1257 |
| $CO_2$ | 759 | 0.4987 | 0.3421 | 0.0897 | 5.9213 | 0.0123 | 0.8134 | 1.9453 |
| T&I | 759 | 0.2335 | 0.1237 | 0.0268 | 1.1923 | 0.0009 | 0.7681 | 3.4452 |
| CF | 759 | 0.1436 | 0.0231 | 0.012 | 0.1498 | 0.0002 | 0.8889 | 2.9432 |

## 4. Results

Table 3 reports the correlation coefficient results among the pairs of applied variables, and the correlation is represented by either a negative sign or a positive sign, as has been shown in the table content. Dormann et al., argue that for the acceptability of the explained variables, the correlation between pairs of variables in the estimated model should not exceed 0.85 [49]. In this context, we can accept the null hypothesis according to the values of the coefficients which do not exceed 0.85, and therefore, the data from this estimation are free from the multicollinearity problem. However, some of the variables show unexpected correlation, but most of them are below 0.5 in coefficient, which shows that these variables do not inherit the problem of simultaneity aside from climate change. Therefore, the results are not biased, and they are fit for policy creation. There is a significant and negative correlation between fossil fuel energy and greenhouse gas emissions and renewable energy. However, the degree of correlation varies. Alternative energy, carbon dioxide gas, innovation technology, and climate change indicate a positive correlation to renewable energy consumption. Moreover, alternative energy, technology innovation, and climate finance show a negative correlation to renewable energy but in the case of carbon dioxide emission gas and greenhouse gas, they show a positive correlation with fossil fuel energy. In the case of fossil fuel, energy shows a negative correlation with renewable gas. Moreover, the finding of the correlation between fossil energy and renewable energy is not significant. Alternative energy production has a significant positive correlation with energy. The results show that the rate of carbon emission produced by alternative energy and renewable energy sources is low compared to greenhouse gas and fossil energy.

**Table 3.** Correlation analysis.

| | RE | FE | AE | $CO_2$ | $CO_2$-GHG | T&I | CF |
|---|---|---|---|---|---|---|---|
| logRE | 1 | | | | | | |
| logFE | −0.534 *** (0.383) | 1 | | | | | |
| logAE | 0.458 *** (−3.013) | −0.123 *** (−3.395) | 1 | | | | |
| $logCO_2$ | 0.561 *** (0.036) | 0.037 *** (1.051) | −0.137 * (0.983) | 1 | | | |
| $logCO_2$-GHG | −0.415 * (−3.23) | 0.0386 *** (0.097) | −0.015 * (4.306) | 0.305 * (−8.9732) | 1 | | |
| log(T&I) | 0.090 *** (2.588) | −0.1243 *** (−3.553) | −0.1787 (−5.1543) | −0.132 *** (−3.7878) | −0.347 *** (−24.120) | 1 | |
| logCF | 0.073 ** (19.85) | −0.165 *** (−4.762) | −0.3699 * (11.2970) | −0.112 *** (3.2330) | −0.481 *** (6959) | 0.488 *** (15.896) | 1 |

Note: *, **, *** represent 10%, 5%, and 1% respectively.

Table 4 reports the pooled OLS regression model, random effect model, and fixed effect model, which show the effect of the proxies of global energy transformation and systematic energy change on climate change and greenhouse gas emissions. Renewable energy, alternative energy, technology and innovation, and financial climate change have a negative effect on climate change. Since the *p*-value associated with the Hausman test is much greater than the conventional threshold of 0.5, we can conclude that the random effects model is more appropriate for this estimation. As the regression output of the random effects model is more appropriate compared to the fixed effect model, this indicates that increasing 1% in renewable energy, alternative energy, technology, and innovation leads to a decrease in the climate change by 0.6959%, 0.51351%, 0.8697%, and 0.413%, respectively. In other words, there is a positive correlation between fossil energy consumption and climate change. The increase in fossil energy consumption increases

climate change by 0.5471%. In the case of greenhouse gas emissions, the system shows there is less emission of greenhouse gas contributed by increasing renewable energy by 0.1234% and alternative energy by 0.1189%. In addition, all indicators have a negative impact in relation to GHG and are statistically significant. In the case of systematic change, a 1% increase in technology & innovation and climate finance negatively impacts greenhouse gas by 0.24326% and 0.09329% and is statistically significant. Fossil energy consumption affects greenhouse gas emissions. It has positively impacted greenhouse gas emissions into the atmosphere by 0.5144%.

**Table 4.** Pooled OLS and random effect models.

| Independent Variables | The Dependent Variable for $CO_2$ (Model 1) | | | The Dependent Variable for GHG-$CO_2$ (Model 2) | | |
|---|---|---|---|---|---|---|
| | Pooled OLS | REM | FEM | Pooled OLS | REM | FEM |
| $logCO_{2(-1)}$ | 1.784 ** | 0.8934 *** | 0.6972 *** | | | |
| | (0.3452) | (0.4321) | (0.7356) | | | |
| $logGHG - COO_{2(-1)}$ | | | | 0.9881 * | 0.8092 *** | 0.7234 *** |
| | | | | (0.7234) | (04373) | (0.6389) |
| Cons | 1.36214 *** | −4.3673 *** | 0.4522 * | −0.342 * | 0.3321 | 0.2349 *** |
| | (3.6069) | (−6.5426) | (12.3421) | (−1.234) | (0.4532) | (0.3872) |
| logRE | −0.6959 *** | −0.1234 ** | 0.2345 | −0.012 ** | −1.213 * | −0.4572 *** |
| | (−13.9701) | (−2.4752) | (0.3452) | (−0.2742) | (−0.214) | (0.2341) |
| logAE | −0.51351 | −0.1189 * | 0.7134 *** | −0.421 *** | −0.137 *** | 0.3636 *** |
| | (−1.2439) | (−0.6236) | (0.8462) | (−0.5611) | (0.3521) | (−0.2379) |
| logFE | 0.5471 | 0.5144 *** | 0.4934 *** | 0.3452 *** | 0.7892 *** | −0.6234 * |
| | (1.4404) | (0.4657) | (0.2234) | (0.8143) | (0.6791) | (0.9023) |
| log(T&I) | −0.0697 *** | −0.04326 ** | 0.4981 ** | −1.234 * | −0.4321 *** | 0.3421 *** |
| | (−4.3412) | (−3.3020) | (0.1983) | (−0.3452) | (−0.2693) | (−0.4871) |
| logFC | −0.4128 ** | −0.9329 *** | 0.7346 * | 0.8956 | −0.8936 *** | 0.2736 ** |
| | (−2.4091) | (−6.4130) | (0.3529) | (0.3425) | (−0.7549) | (0.3342) |
| R-Squared | 0.8869 | 0.9333 | 0.7945 | 0.8179 | 0.9128 | 0.9184 |
| F-Statistic | 191.3245 | 337.26 | 145.92 | 112.7845 | 79.564 | 134.992 |
| Prob(F-Static) | 0.0405 | 0.0230 | 0.000 | (0.000) | (0.0012) | 0.000 |
| Hansen test | | | 74.256 | | | 23.47 |
| Prob (F-Statistic) | | | 0.3432 | | | 0.3946 |

Note: *, **, *** represent 10%, 5%, and 1% respectively.

The GMM estimator has been used to control the potential of endogeneity of all explanatory variables cited from Arellano and Bond, as they popularized the work of Holtz-Eakin et al. [50]. Thus, the lagged differences of the regressors can be used as additional instruments for a level equation. Moreover, the consistency of the GMM estimator is based on the two specification tests, that is the Hansen test and the serial correlation (or autocorrelation) test [51]. According to the Hansen test, namely over-identifying restriction, the failure to reject the null hypothesis would imply that the instrument variables are valid and the model is correctly specified. In the case of the serial correlation test, the test of no first and second-order serial correlations in the residuals of the first-difference equation is determined. The rejection of the null hypothesis due to the absence of the first-order serial correlation AR (1) and failure to reject the absence of the second-order serial correlation AR (2) will ensure and conclude that the applied model is correctly specified [52]. In this study, the sys-GMM models consist of the main regression model and robustness check-up for the models, and each contains the proxy variables concerning either global transformation or systematic change in climate change and greenhouse gas emissions.

Table 5, system models 1 and 2 represent the regressions of systematic energy change of both climate change and greenhouse gas emissions, respectively. Technology innovation and financial climate are kept as instrumented variables, while renewable energy, alternative energy, and fossil energy are control variables. As far as the lagged dependent on climate change variables and greenhouse gas is concerned, there is persistence in climate change; the previous year's climate change and greenhouse gas emissions are a predictor of the current year's climate change and greenhouse gas emissions.

**Table 5.** Sys-generalized method of momentum with robustness regression check-up (climate change and greenhouse gaseous are independent variables).

| | **Main Regression System** | | | |
|---|---|---|---|---|
| | **(1)** | **(2)** | **(3)** | **(4)** |
| | **($CO_2$)** | **(GHG_$CO_2$)** | **($CO_2$)** | **(GHG_$CO_2$)** |
| $\log CO_{2(t-1)}$ | 0.5927 *** | | 0.5919 *** | |
| | (40.0724) | | (6.6379) | |
| $\log(GHG\_CO_{2(t-1)})$ | | 0.6762 ** | | −1.2281 * |
| | | (0.8461) | | (−1.8127) |
| logRE | −0.1548 *** | −0.0644 *** | 0.2641 *** | 0.09023 *** |
| | (−15.5684) | (−0.8509) | (1.6562) | (1.6459) |
| logAE | −0.3456 *** | −0.4556 *** | −0.56432 *** | −0.3456 *** |
| | (−6.4321) | (−0.6745) | (0.12467) | (0.4321) |
| logFE | 0.7485 *** | 0.06731 *** | 0.2449 *** | 0.03069 |
| | (7.5730) | (0.6766) | (1.6552) | (1.5890) |
| log(T&I) | −0.2567 *** | −0.5621 *** | −0.2245 *** | −0.7213 ** |
| | (−0.345) | (0.4128) | (0.5432) | (0.3467) |
| logCF | 0.34256 | 0.8213 | 0.6532 | 0.74531 |
| | (0.5421) | (0.4916) | (0.1124) | (0.9352) |
| Year Dummies | Yes | Yes | Yes | Yes |
| No of Observation | 719 | 679 | 719 | 659 |
| Hansen ρ values | 0.2765 | 0.3214 | 0.3452 | 0.1245 |
| | (0.1234) | (0.3421) | (0.3321) | (0.3412) |
| AR(1) *p*-values | 0.4231 *** | 0.3723 ** | | |
| | (0.6123) | (0.3546) | | |
| AR (2) ρ values | 0.3214 | 0.4592 | | |
| | (0.7562) | (0.5632) | | |

Note: *, **, *** represent 10%, 5%, and 1% respectively.

In the global energy transformation change, the results indicate that renewable energy affects both climate change and greenhouse gas emissions. Furthermore, the outputs are statistically significant and yield negative effects at a 1% level. The empirical analysis shows that a 1% increase in renewable energy produced leads to 0.155684% and 0.0644% decreases in climate change and greenhouse gas emissions, respectively. In another case, alternative energy has a negative impact on climate change and greenhouse gas emissions and is statistically significant at a 1% level; a 1% increase in alternative energy produced leads to 0.3456% and 0.04556% decreases in climate change and greenhouse gas emissions, respectively. Alternatively, fossil energy production has a positive and statistically significant impact on climate change and greenhouse gas at a 1% level. About a 1% increase in fossil energy consumption leads to 0.7485% and 0.06731% increases in climate change and greenhouse gas, respectively.

Also, Table 5 indicates the effects of global transformation energy change and systematic energy change on climate change. The systematic change includes technology and innovation and finance climate on both climate change and greenhouse gas emission.



In this analysis, renewable energy, and alternative energy, are kept as the instrument variables, and technology and innovation, as well as climate finance, are control variables. Technology and innovation shows a negative effect on climate change and is statistically significant at a 1% level. It is about 0.4031% and 0.1003% for climate change and greenhouse gas emissions, respectively. Despite innovation and technology, climate finance shows a statistically insignificant impact on both climate change and greenhouse gas. Seemingly, the finance climate has a lower percentage, and it is insignificant due to the fact that the funds that have been allocated for climate change mitigation need to be better satisfied.

The robustness check-up has added equivalence results in this model to ensure the credibility of sys-GMM results. All in all, the empirical results indicate the efficiency and significance of robustness output as revealed from the main sys-GMM.

## 5. Discussion

The system models represent the regressions of global transformation and systematic energy change of both climate change and greenhouse gas emissions. In this study, the empirical analysis shows that renewable energy, alternative energy, technology and innovation, and financial climate have a negative effect on climate change. In the global energy transformation change, the results indicate that renewable energy affects both climate change and greenhouse gas emissions. The outcome is supported by the research conducted by Olabi et al., in which they found that clean renewable energy confines the emission of greenhouse gases and current progress in climate change [53]. Chein et al., justify the results that the consumption of renewable energy mitigates greenhouse gas [54]. Sun et al., justify that renewable energy consumption and seemingly green innovation abate carbon emissions globally. This is significant because increasing consumption of the transformation energies leads to reducing the effect of climate change. In another case, alternative energy has a negative and statistically significant impact on climate change and greenhouse gas emissions. It means that an increase in alternative energy production leads to a decrease in climate change and greenhouse, respectively. The comparative analysis regarding strengths, weaknesses, opportunities, and threats justifies that power plants are not only limited to human health or noise pollution but are also actively being used to reduce greenhouse emissions, depletion of the ozone layer, eutrophication, dried-up rivers, toxication, and flooding [55]. Alternatively, fossil energy production has a positive and statistically significant impact on climate change and greenhouse gases. An increases in fossil energy consumption leads to an increase in climate change and greenhouse gas, respectively. The research shows that fossil energy for industrial consumption contributes to the emission of a large percentage of carbon dioxide and greenhouse gas as well. The results revealed that the global energy transformation policy has a tremendous positive impact on climate change mitigation. Resai et al., reported that any increase in the parameter of fossil energy will result in climate change, and it has been discovered that the crucial factor in carbon dioxide emission is crude oil consumption, which has an immense effect on climate change mitigation [56]. After all, the global transformation energy policy minimizes the effect of climate change as well as lowering the effect of global warming, which is already affecting the world's human and natural systems. Since the global transformation consists of renewable energy and alternative energy that have a low rate of producing an excess of carbon dioxide gas into the atmosphere, the low rate of carbon dioxide emission minimizes the rate of climate change. The effect of fossil fuel energy consumption on climate change is very high compared to renewable energy and alternative energy. This shows that industries' fossil fuel energy consumption is very high. Fossil fuel energy produces a higher rate of carbon in the atmosphere, which significantly increases the rate of climate change on this planet. The consumption of fossil fuel energy results in greenhouse gas production and carbonization of the atmosphere. Systematic change encompasses climate finance through technology and innovation, both of which

determine the impact of climate change and greenhouse gas emissions. Innovation and technology show a negative effect on climate change which is statistically significant at a 1% level. The findings revealed that increases in technology and innovation in producing clean energy diminish the rate of climate change. Technology and innovation play a significant role in climate change mitigation. Therefore, in order to mitigate climate change globally, there is a need to invest more in the area of technology and investment so as to have well-advanced technology just to filter the effect of carbon dioxide and greenhouse gas production in the atmosphere that contributes to climate change. Despite innovation and technology, finance climate shows a statistically insignificant effect on both climate change and greenhouse gas. Improving the finance climate for funding all the projects related to green environments helps to minimize the effect of carbon monoxide, carbon dioxide, and greenhouse gas emissions that reduce the effect of climate change. Seemingly, climate finance is revealed to have a lower percentage and statistical insignificance due to the fact that the budget for climate change mitigation is not well-satisfied. Therefore, to overcome the effect of climate change and greenhouse gas emissions, global climate finance funds should be improved.

## 6. Conclusions

This study has been conducted to evaluate the impact of global energy change and systematic energy change on climate change mitigation. The study used dynamic panel data which covers the comprehensive dataset from World Development Indicators, International Atomic Energy Agency, and International Monetary Fund with 26 world regions over the period from 2005 to 2022. This study increases the popularity of lowering the global temperature which will lead to reducing the effect of climate change within our regions. As the literature indicates, as greenhouse gas (GHG) emissions continue overwhelmingly, the rate of global warming gradually increases. The increase in global warming by 1.5 °C causes global multiplier effects, such as melting glaciers in the Antarctic, the melting of Greenland ice sheets, a higher frequency of severe hurricanes, a greater severity of droughts, and forest fires on the Earth. These catastrophes and calamities have been caused by climate change. To come up with estimated results, the study uses the proxies of global transformation: renewable energy, fossil energy, and alternative energy. In systematic change, the investigation uses the proxies of technology and innovation as well as finance climate. Both global transformation and systematic change have been used as global policies for climate change mitigation. The study employed a quantitative approach that applies pooled OLS regression model and system Generalized Moments of Method (sys-GMM). For a safe climate, this study examined the effects of key policies of global transformation and systematic change on climate change. The results indicate that on the one hand, renewable energy has a negative impact on climate change, and it is statistically significant at a 1% percentage level. This means that a 1% increase in the share of renewable energy consumption in the total energy use will result in a reduction of approximately 0.1548% in $CO_2$ emissions from gas fuel consumption that contributes to climate change. In addition, increasing the consumption of renewable energy reduces the effect of climate change. The outcome of the result continues to emphasize a greater consumption of renewable energy to mitigate the adverse effects of climate change as well as global warming. To achieve a net-zero temperature, as has been agreed at the Glasgow summit, the world should work and cooperate by stimulating the rate of using renewable energy as well as clean energy from industries to domestic consumption. In fact, renewable energy produces less carbon dioxide and greenhouse gas, preventing the Earth from being harmed by global warming.

However, fossil energy is statistically significant and positively affects climate change. Increasing the consumption of fossil energy raises the effect of climate change. The global need for massive decarbonization infrastructure will help minimize global warming that leads to climate change. Policies that take an endogenous approach through global transformation and systematic change should be implemented to reduce the effect

of climate change. The policy should reduce the consumption of non-renewable energy and increase the consumption of renewable energy. Furthermore, the energy sector is developing in Asia but not necessarily in other continents such as, for example, Europe [57]. It is of great significance for the European economy due to predictions that global energy demand will be increased but not as much in the European zone [58].

On the other hand, alternative energy has a negative effect on climate change and is statistically significant at a 1% level. It means that 1% of alternative energy in total energy use reduces approximately 0.1548% of the $CO_2$ emissions from gas fuel consumption that contributes to climate change. It shows that increasing the rate of alternative energy consumption minimizes the effect of climate change, although the rate of consumption of alternative energy is lower compared to the rate of renewable energy. More effort should be made to improve the consumption of alternative energy so as to reduce the rate of effect of climate change globally. However, fossil energy production has a positive effect on climate change and is statistically significant at a 1% level. It means that 1% of the total energy use of fossil energy led to an increase of about 0.7485% of $CO_2$ emissions from gas fuel consumption, contributing to climate change. The finding revealed that the rate of consumption of fossil energy is the highest compared to renewable energy and alternative energy. The continuing increase in the consumption of fossil energy sources in the production areas as the global energy source leads to an increase in the effect of climate change. The uses of non-renewable energy produce access to carbon dioxide and greenhouse gas (GHG), which increase the global temperature that leads to the creation of climate change.

In the case of technology and innovation, the proxies of systematic change have a negative effect on climate change that is statistically significant at the 1% level. It means that 1% of the total energy use of technology and innovation led to a decrease about 0.2567% of $CO_2$ emissions from gas fuel consumption that contributes to climate change. Furthermore, technology has significantly contributed to mitigating the effects of climate change on the Earth. Therefore, the filter that removes the carbon dioxide gas and greenhouse gases should be implemented in the vast production industries that can prevent climate change. The technology of constructing green buildings and planting trees absorbs the carbon dioxide gas produced into the atmosphere and plays a significant part in climate change mitigation. Climate finance shows less contribution to climate change mitigation. It represents inadequate funds to govern all the projects related to green environments for decarbonization and greenhouse gas emissions for climate change mitigation. Although, it is profound that the climate finance policy deliberates the climate change mitigation initiative for environmental protection by implementing clean energy investments so that individuals and collectives can change their behavior and maintain the possibilities of climate change mitigation [16]. However, various scientific researchers related to climate change conducted meetings on climate change and conferences with prestigious scholars that include climate finance institutions who gathered to find the solutions for reducing carbon dioxide and greenhouse gas emissions, but unfortunately, global warming is still accelerating as time elapses.

Therefore, global transformation and systematic change, once effectively implemented and maintained, will reduce the effect of climate change. Global transformation and systematic change policies have a tremendous positive impact on climate change mitigation. After all, global transformation and systematic change policies minimize the effect of climate change as well as lowering the effect of global warming, which is already affecting the world's human and natural systems. The global energy consumption requires a massive investment in climate change mitigation policy through climate finance funding projects for a decarbonization infrastructure that leads to minimizing the effect of global warming and climate change. The global need for massive decarbonization infrastructure will help minimize global warming, which leads to climate change. Endogenous approach policies through global transformation and systematic change should be implemented to reduce the effect of climate change. The policy should

reduce the consumption of non-renewable energy and increase the consumption of renewable energy.

**Author Contributions:** Conceptualization, H.G., H.M.A.S., H.K. and M.Ş.; methodology, H.G. and H.M.A.S.; software, H.G. and H.K.; validation, H.G., H.M.A.S. and H.K.; formal analysis, H.G. and H.M.A.S.; investigation, H.G., H.M.A.S., H.K. and M.Ş.; resources, H.G. and H.M.A.S.; data curation, H.M.A.S.; writing—original draft preparation, H.G., H.M.A.S., H.K. and M.Ş.; writing—review and editing, H.G., H.K. and M.Ş.; visualization, H.G.; supervision, H.G. and H.K. All authors have read and agreed to the published version of the manuscript.

**Funding:** This research received no external funding.

**Institutional Review Board Statement:** Not applicable.

**Informed Consent Statement:** Not applicable.

**Data Availability Statement:** Not applicable.

**Conflicts of Interest:** The authors declare no conflict of interest.

## Appendix A

**Table A1.** The World Regions.

| | | |
|---|---|---|
| Western and Central Africa | Arab World | Central Europe and the Baltics |
| East Asia and Pacific | East Asia and Pacific (excluding high-income) | East Asia and Pacific (IDA and IBRD countries) |
| Europe and Central Asia (IDA and IBRD countries) | European Union | Latin America and the Caribbean |
| Latin America and the Caribbean (IDA and IBRD countries) | Least developed countries: UN classification | Low and middle income |
| Low income | Middle East and North Africa | Middle East and North Africa (excluding high income) |
| Middle East and North Africa (IDA and IBRD countries) | Middle income | North America |
| North Macedonia | OECD members | Small states |
| Sub-Saharan Africa | Sub-Saharan Africa (excluding high-income) | Sub-Saharan Africa (IDA and IBRD countries) |
| Upper middle income | | |

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
