# Peer review of "Global Energy Transformation and the Impacts of Systematic Energy Change Policy on Climate Change Mitigation"

_sustainability, doi:10.3390/su151914298_

Round 1

Reviewer 1 Report

Global Energy Transformation And Systematic Energy Change Policy On Climatic Change Mitigation

the paper provides important insights into the impact of different energy sources, including renewable energy, alternative energy, and fossil energy, on climate change. The authors effectively highlight the negative effect of consumption of non-renewable energy on climate change, while emphasizing the positive impact of renewable energy and innovation in mitigating its effects. This nuanced analysis contributes to our understanding of the complexities involved in addressing climate change. It contributes to the existing literature on climate change and energy transition, and their policy recommendations can potentially inform strategies for sustainable development and environmental conservation. However, I have some suggestions to improve the paper:

Begin the paper by clearly stating the objective of the study, such as evaluating the impact of global energy transformation and systematic energy change on climate change. This will provide readers with a clear understanding of the research focus. And provide a brief overview of the methodology: Instead of listing the data sources and techniques used without much context, briefly explain the methodology employed. For example, mention that dynamic panel data analysis techniques, such as Generalized Method of Moments (GMM), Pooled OLS, and Random Effects models were used to empirically evaluate the linked effect of global transformation and systematic change on climatic change. Also, Clarify the rationale for using the system-GMM approach: Explain why the system-GMM approach was chosen and its advantages in controlling for endogeneity, omitted variable bias, measurement errors, and unobserved panel heterogeneity. Instead of listing the findings, consider presenting them in a clearer and more structured way. Use subheadings or bullet points to highlight the key empirical results, making it easier for readers to grasp the main findings. Elaborate on the findings related to the effects of renewable energy, alternative energy, technology & innovation, financial climate, and fossil energy on climate change. Explain the statistical significance, magnitude, and implications of these effects. Additionally, consider discussing potential reasons for the observed relationships, drawing on relevant literature or theories. After presenting the empirical results, discuss the policy implications based on the findings. Provide concrete recommendations on how global transformation and systematic change can be used to reduce the effect of climate change. Elaborate on the importance of decarbonisation infrastructure and the potential benefits of reducing the consumption of non-renewable energy while increasing the consumption of renewable energy.

a careful proofread would be great to improve the paper 

Author Response

Dear Reviewers,

We have revised the article. Based on suggestions. Thank you for your contributions.

Best regards

Reviewer 2 Report

Thank tou for reviewing possibility of this interesting paper, however i have few suggestions that might improve the paper:

- The goal of the paper doesn't flow from previous studies. Also authors should state what new they introduce to existing knowledge

- Line 89: COP 26 Glasgow (not Gloscow)

- Chapter contains both theoretical background and materials and methods. It needs to be divided into two chapters. In theoretical backround i suggest to add some words that energy sector is increasing in Asia, but not necessarily in other countrieslike European (https://doi.org/10.3390/en14227579). It’s important for european economy in case of predictions that energy demand in the world will be increased, but not much in Europe (https://doi.org/10.3390/resources8020100).

- Line 359: Authors study 26 world regions. they should stste why those regions have been chosen. What part of the world they cover (whole?)

- Chapter 4 doesn't contains discussion. In this part authors should show there their main results and discuss them with previous studies. 

Author Response

(The authors gave the same response as above.)

Round 2

Reviewer 1 Report

thank you for considering my suggestions. it would be better if you get the high resolution for the figure in order t make it clearer in the published article  

Author Response

Dear Refereer;

We wish to submit an original research article entitled “Global Energy Transformation and Systematic Energy Change Policy On Climatic Change Mitigation” for consideration by the MDPI Sustainability Journal.  We confirm that this work is original and has not been published elsewhere, nor is it currently under consideration for publication elsewhere. In this paper, the empirical analysis shows that renewable energy, alternative energy, technology & innovation, and financial climate have a negative effect on climate change. This is significant because increasing consumption of the transformation energies leads to reducing the effect of climate change. However, fossil energy is statistically significant and positively affects climate change. Increasing consumption of fossil energy raises the effect of climate change. The parameter of climate finance is not efficient means that more effort should do so as to increase the rate of climate finance. The global need for massive decarbonization infrastructure will help minimize global warming that leads to climate change.   We believe that this manuscript is appropriate for publication in MDPI Sustainability Journal because the aim of MDPI is to foster scientific exchange in all forms, especially the global energy policy across all disciplines. MDPI's guidelines for disseminating open science. As the aim of this study is to evaluate the effect of global energy transformation and systematic energy change on climate change

The global need for massive decarbonization infrastructure will help minimize global warming that leads to climate change. Endogenous approach policies through global transformation and systematic change should be implemented to reduce the effect of climate change. The policy should reduce the consumption of non-renewable energy and increase the consumption of renewable energy.

We have no conflicts of interest to disclose.

Reviewer 2 Report

Authors did not followed my suggestions:

- chapter Materials and methods has not been upgraded with my suggestions and suggested references (see my comments in first review)

- Discussion and Conclusion chapter is not improwed correctly. There is still no discussion in the text (see comments in my first review).

Those suggestions must be included before acceptance of the paper.

Author Response

(The authors gave the same response as above.)

Round 3

Reviewer 2 Report

Authors did not followed my suggestions. Submitted paper is the same as previous version. I'm confused about authors response for my second review. It contains plenty of words about the paper, but not refer to my suggestions. If authors not include my all suggestions from 1st and 2nd review I'll recommend rejection of the paper.

Author Response

Dear Referee;

Our article has been revised per your 1st and 2nd revision suggestions. Your suggestions have been cited. Thank you very much for your tips to improve the quality of our article. We are waiting for your reply to review our article to see if we missed anything.

Round 1

Comment 1: - The paper's goal doesn't flow from previous studies. Also, authors should state what new they introduce to existing knowledge

Response from comment 1: The goal of this study is to evaluate the effect of global energy transformation and systematic energy change on climate change mitigation. The reviewed literature from different papers indicates that most of the thoughts discussed climate change in terms of renewable energy or nonrenewable energy but the flows of this paper discuss climate change change mitigation in the context of global energy transformation and systematic energy change.

The new introduced to existing knowledge is to investigate the effect of climate change on global transformation energy on the proxy variable of renewable energy, nonrenewable energy, and alternative energy. The systematic change contains the proxies of technology & innovation and the financial climate

Comment 2: - Line 89: COP 26 Glasgow (not Gloscow)

Response:  Line 89: COP 26 Glasgow (not Gloscow)

Yes, the reference has been done by correcting line 89 by writing COP 26 Glasgow instead of COP 26 Gloscow.

Comment 3: - Chapter contains both theoretical background and materials and methods. It needs to be divided into two chapters. In theoretical backround i suggest to add some words that energy sector is increasing in Asia, but not necessarily in other countrieslike European (https://doi.org/10.3390/en14227579). It’s important for european economy in case of predictions that energy demand in the world will be increased, but not much in Europe (https://doi.org/10.3390/resources8020100).

Response:  The referencing of the chapters: The correction of the chapter has been made by separating the chapters has been made by introducing the Theoretical background and Materials and methods. Thus, the corrections have been done to divide the chapter into two chapters. In the theoretical background, we made as has been suggested should add the words” energy sector is increasing in Asia, but not necessarily in other countries like European “referencing (https://doi.org/10.3390/en14227579). And “it’s important for the European economy in case of predictions that energy demand in the world will be increased, but not much in Europe” referencing (https://doi.org/10.3390/resources8020100).

Comment 4: - Line 359: Authors study 26 world regions. they should stste why those regions have been chosen. What part of the world they cover (whole?)

Response: . The reason for choosing these 26 regions is to assess the impact of energy conversion and systematic energy change to mitigate climate change on a global scale. Due to the difficulty of accessing the data quality related to climate change, this study investigates the effect of climate change by using balanced panel data using the 26 world economic regions. The data presentation for investigating the global effect of climate change is based on the demographic and production rate of climate change and greenhouse gas emissions in the atmosphere.

Comment 5: - Chapter 4 doesn't contain a discussion. In this part, authors should show there their main results and discuss them with previous studies. 

Response:  The corrections have been made for Chapter 4 which contains the discussions section. This part by indicating the main results and discuss them with previous studies.

Round 2

Comment 1: - chapter Materials and methods has not been upgraded with my suggestions and suggested references (see my comments in first review)

Response: The chapter Materials and methods has been upgladed by explaining the aim of the study is to evaluate the effect of global energy transformation and systematic energy change on climate change. Also, it has been upgrdaded by depecting the data that constructed from balanced dynamic panel data, which comprises 26 world regions as represented in Appendix A from the Database Indicators (WDI), International Energy Atomic (IEA), and International Monetary Fund (IMF), which spans from 2005 to 2022. Further the reasons of choosing these 26 regions is to assess the impact of energy conversion and systematic energy change to mitigate climate change on a global scale. Due to the difficulty of accessing the data quality related to climate change, this study investigates the effect of climate change by using balanced panel data using the 26 world economic regions. The data presentation for investigating the global effect of climate change is based on the demographic and production rate of climate change and greenhouse gas emissions in the atmosphere.

Comment 2: - Discussion and Conclusion chapter is not improwed correctly. There is still no discussion in the text (see comments in my first review).

Response:

  1. the section has been upgraded by discussing the finding results of existing study by comparing the similar results with the previous studies as has been presented. Therfore, the discussion from the thoughts and arguments are related from the existing study and previous investigations.
  2. The conclusion section explains the findings and the policy recommandation from the output as has been obtained from the emperical analysis. The conclusion section highlights the policy recommnadton that The global need for massive decarbonization infrastructure will help minimize global warming that leads to climate change. Endogenous approach policies through global transformation and systematic change should be implemented to reduce the effect of climate change. The policy should reduce the consumption of non-renewable energy and increase the consumption of renewable energy.

Round 4

Reviewer 2 Report

I accept paper in present form